

# Identification of single nucleotide polymorphisms (SNPs) potentially associated with residual feed intake in Qinchuan beef cattle by hypothalamus and duodenum RNA-Seq data

Zonghua Su[1], Chenglong Li[1], Chaoyun Yang[1], YanLing Ding[1], Xiaonan Zhou[1], Junjie Xu[1], Chang Qu[1], Yuangang Shi[1,†], Cong-Jun Li[2] and Xiaolong Kang[1]

[1] Key Laboratory of Ruminant Molecular and Cellular Breeding, College of Animal Science and Technology, Ningxia University, Yinchuan, China
[2] Animal Genomics and Improvement Laboratory, Henry A. Wallace Beltsville Agricultural Research Center, Agricultural Research Service, United States Department of Agricultural, Beltsville, MD, United States
[†] Deceased author.

Corresponding authors
Cong-Jun Li, congjun.li@usda.gov
Xiaolong Kang, kangxl9527@126.com

## ABSTRACT

The regulation of residual feed intake (RFI) in beef cattle involves brain-gut mechanisms due to the interaction between neural signals in the brain and hunger or satiety in the gut. RNA-Seq data contain an extensive resource of untapped SNPs. Therefore, hypothalamic and duodenal tissues from ten extreme RFI individuals were collected, and transcriptome sequenced in this study. All the alignment data were combined according to RFI, and the SNPs in the same group were identified. A total of 270,410 SNPs were found in the high RFI group, and 255,120 SNPs were found in the low RFI group. Most SNPs were detected in the intronic region, followed by the intergenic region, and the exon region accounts for 1.11% and 1.38% in the high and low RFI groups, respectively. Prediction of high-impact SNPs and annotation of the genes in which they are located yielded 83 and 97 genes in the high-RFI and low-RFI groups, respectively. GO enrichment analysis of these genes revealed multiple NADH/NADPH-related pathways, with *ND4*, *ND5*, and *ND6* significantly enriched as core subunits of NADH dehydrogenase (complex I), and is closely related to mitochondrial function. KEGG enrichment analysis of *ND4*, *ND5*, and *ND6* genes was enriched in the thermogenic pathway. Multiple genes, such as *ATP1A2*, *SLC9A4*, and *PLA2G5*, were reported to be associated with RFI energy metabolism in the concurrent enrichment analysis. Protein-protein interaction analysis identified multiple potential candidate genes related to energy metabolism that were hypothesized to be potentially associated with the RFI phenotype. The results of this study will help to increase our understanding of identifying SNPs with significant genetic effects and their potential biological functions.

## INTRODUCTION

Feed cost accounts more than 70% of the total input cost in cattle production, making feed utilization a crucial metric for evaluating production expenditures (*Patience, Rossoni-Serao & Gutierrez, 2015*). Efficient feed utilization can reduce herd maintenance costs by 9–10%, lower feed intake by 10–12%, and decrease methane emissions by 15–20% (*Moore, Mujibi & Sherman, 2009*). Thus, optimizing feed utilization and reducing production costs are essential for livestock development. Residual feed intake (RFI) represents the disparity between the average daily feed intake (ADFI) and the average expected feed intake (AEFI) needed to maintain production levels (*Koch et al., 1963*). RFI provides a precise measure of feed utilization efficiency in livestock, isolating the effects of animal growth traits and rates (*Richardson & Herd, 2004*).

RFI is a promising candidate for genetic improvement due to its moderate heritability (0.28–0.58) (*Moore, Mujibi & Sherman, 2009*) and significant genetic variability (*Archer & Bergh, 2000*; *Herd & Bishop, 2000*). Our research found that RFI is related to gut microbiota (*Zhou et al., 2023*), circRNA-miRNA interaction (*Zhao et al., 2023*), and gene expression (*Yang et al., 2023*, *2021*, *2022*). These studies support a comprehensive analysis of RFI and show that its influencing factors are numerous and complex.

The hypothalamus and duodenum are critical organs in animal feed intake, energy metabolism, and digestion. The hypothalamic arcuate nucleus regulates appetite, where neuropeptide Y (NPY) and agouti-related peptide (AGRP) promote feeding. Conversely, α-MSH (α-melanocyte-stimulating hormone) induces satiety (*Perkins et al., 2014*). The duodenum, a key organ for nutrient absorption, facilitates various metabolic functions such as glucose, fat, vitamin B, calcium, zinc, and iron (*Anand et al., 2021*; *Cooke & Clark, 1976*; *Reeves & Chaney, 2004*). The interplay between the central nervous and digestive systems is evident in the microbiota-gut-brain axis (MGBA). The nervous system influences gut function through neurotransmitters and gut hormones, while gut microbes play crucial roles in host nutrient metabolism (*The 1000 Genomes Project Consortium, 2015*; *Olivier, 2003*). Thus, the close association of the hypothalamus and duodenum with feeding efficiency underscores their importance in studies investigating RFI in beef cattle.

Single nucleotide polymorphism (SNP) is a genetic variation resulting from a single nucleotide change in the DNA sequence. These variants can influence phenotypes and disease susceptibilities (*Kim & Misra, 2007*). Due to the low cost and high availability of RNA-seq data, coding region variants from RNA-Seq data are widely studied for their potential contribution to phenotype (*Karczewski et al., 2020*). Transcriptome data offer gene expression levels that can be utilized to investigate cis-regulation based on the expression of genes with SNP sites (*Jehl et al., 2021*). A wealth of research has been dedicated to extracting SNPs from transcriptomic data, yielding significant advancements across various fields. For instance, transcriptome sequencing of cow's milk has facilitated the discovery of SNPs, providing a robust foundation for marker-trait association studies (*Canovas et al., 2010*). In aquaculture, RNA-seq analysis has identified SNPs potentially linked to the immune response and the growth performance of *Penaeus vannamei* (*Santos, Andrade & Freitas, 2018*). In crop science, the development of genome-wide SNP markers

for barley has been achieved through reference-based RNA-Seq analysis (*Tanaka et al., 2019*). In animal husbandry, RNA-Seq SNP data has revealed potential causal mutations relevant to pig production traits and the intricacies of RNA editing (*Martinez-Montes et al., 2017*). SNP analysis of transcriptomic data from 20 human and bovine tissues revealed that cis-regulatory elements of gene expression are conserved between humans and cattle (*Yao et al., 2022*). Differential expression of an intronic SNP in *FABP4* was found to correlate with lipid transport and intracellular homeostatic regulation in studies of bovine rumen acidosis (*Zhao et al., 2017*).

This study characterized the SNPs from the hypothalamus and duodenum tissues of the same cattle with high and low residual feed intake based on RNA-seq data. The objective was to identify SNPs related to beef RFI and conduct subsequent bioinformatics analysis to detect the functional SNPs/genes associated with feed utilization performance in beef cattle and expand our understanding of the role of genetic variants in RFI phenotypes from expressed regions of the genome.

## MATERIALS AND METHODS

### Experimental animals and data collection

Based on our previous study (*Yang et al., 2021*), 30 Qinchuan bulls with similar age ($15 \pm 1$ months) and weight ($280.6 \pm 30.9$ kg) were selected from a farm in Ningxia, China. The study subjects were given a standardized feeding regimen throughout the experimental period, and free access to water and food was ensured. Body weight measurements were taken monthly throughout the 81-day experimental period, then daily feed intake, average daily gain (ADG), and the midpoint metabolic body weight (MMBW0.75) was calculated based on feed intake (FI). To classify the cattle into high and low RFI groups, we used multiple linear regression of FI on the midpoint MMBW^0.75 and ADG to estimate individual RFI (*Yang et al., 2021*).

### RNA extraction and sequencing

Based on the results of the RFI calculation, five individuals with extremely low RFI (LRFI, high efficiency) and high RFI (HRFI, low efficiency) phenotypes were selected for slaughter after a 16-h fasting period. All experimental procedures involving animals were conducted by the Guidelines for Ethical Review of Laboratory Animal Welfare of Ningxia University (NXUC20211015). The hypothalamus (including the arcuate strong nucleus, parabrachial nucleus, supraoptic nucleus, dorsal/ventral medial nucleus, and other brain tissues) and the descending duodenum (mucosa, submucosa, and external muscular propria) were collected post-slaughter. Our tissue samples include hypothalamus and duodenum tissues from five high RFI and five low RFI cattle, totaling 20 samples. These tissue samples were washed with PBS to remove blood and other impurities, then cut into small pieces to increase surface area, and placed into sterile tubes for further processing. Total RNA was extracted from 500 mg of tissue samples using TRIzol method (TaKaRa Bio, Beijing, China), following the manufacturer's instructions. After extraction, the RNA was further purified using column purification to enhance purity and remove residual contaminants.

DNase treatment was performed to eliminate genomic DNA contamination. The quality and integrity of the extracted RNA were assessed using 1% agarose gel electrophoresis, Nanodrop, and Agilent 2100 to ensure a sample concentration of ≥500 ng/μL, 28S:18S > 1.0, and RIN ≥ 7. For library construction, the RNA was first reverse transcribed into cDNA. Adapter ligation was then performed to facilitate the sequencing process, followed by amplification to enrich the library. The library's initial quantification was carried out using Qubit 2.0, and the insert size was verified using an Agilent 2100. The effective concentration of the library (effective concentration > 2 nM) was accurately determined using qRT-PCR. The hypothalamus tissue samples underwent whole transcriptome sequencing to capture both coding and non-coding RNA species, providing a comprehensive view of the transcriptome. The duodenum tissue samples were subjected to standard transcriptome sequencing. Finally, pair-end sequencing data (raw data) with 150 bp read length were generated using the Illumina HiSeq 4000 platform. High RFI sequencing data of hypothalamus and duodenum were named Q_H1~Q_H5 and S_H1~S_H5, respectively, while low RFI sequencing data were named Q_L1~Q_L5 and S_L1~S_L5, respectively.

## Quality control, mapping and transcript assembly

The statistical power of this experimental design, calculated in RNASeqPower (https://bioconductor.org/packages/release/bioc/html/RNASeqPower.html) is 0.86 (The corresponding code can be found in Script S1). The quality of raw data was assessed using the fastQC v.0.11.9. Subsequently, the Trimmomatic v.0.39 was used to perform quality control on the data. This included removing adapter sequences, trimming bases with Phred scores below 30 at the beginning and end of reads, applying a sliding window approach with a window size of 5 bp to remove bases with an average Phred score below 20, and discarding reads shorter than 75 bp. The cleaned data were then reevaluated using the fastQC software to ensure they met the requirements for subsequent analysis. The clean data were aligned to the bovine reference genome (ARS-UCD1.2; INSDC Assembly) using the STAR v.2.7.3a with the following parameters: –outSAMtype BAM Unsorted SortedByCoordinate, -outFilterMismatchNmax 999, –outFilterMismatchNoverReadLmax 0.04, –outFilterMultimapNmax 1. The resulting alignment files were further processed using the AddOrReplaceReadGroups tool of the PICARD v.2.27.4. This added the sample ReadGroups (RG) information to each alignment file. Additionally, the MarkDuplicates tool was applied to remove duplicate amplifications resulting from the PCR process during library construction.

## Merging of sample data

For increasing the number of reads per variant locus, enhancing the depth coverage of reads across the entire transcriptome, as well as the depth coverage and quality of variant calls (Lam et al., 2020), in this study, the data from hypothalamic and duodenal tissues were merged into two BAM files based on phenotype (high RFI group and low RFI group). This merging process, performed using the "merge" command of the samtools v.1.16.1,

aimed to balance sequencing depth between samples and minimize the impact on SNP analysis results. Both the high RFI and low RFI groups in the subsequent analysis referred to the combined group data (Fig. 1).

## SNPs recognition, filtering and annotation

BCFtools v.1.16 was utilized to execute the variant calling on the combined data of the high RFI group and low RFI group respectively, enabling identification of SNP sites and generating BCF files containing variant information. The "norm" parameter of BCFtools was then employed to normalize the variant information, thereby eliminating ambiguity caused by varying methods. Subsequently, the low-quality SNPs data underwent further filtering to reduce the likelihood of false positives and alleviate computational resource requirements for subsequent analysis. The software BCFtools and VCFtools v.0.1.16) were employed for variant filtering, employing the following criteria: (1) Removal of SNPs within a 5 bp range near indels; (2) setting a minimum coverage (DP) of 10; (3) enforcing a minimum allele frequency not less than 0.2 and a secondary allele depth not less than 2; (4) filtering loci with quality scores below 30. Finally, the functional annotation of SNPs was performed using the SnpEff v.5.1d with the built-in ARS-UCD1.2.105 database. The thresholds of above software are referenced from previous study (*Lam et al., 2020*).

## Identifying and annotating high and low RFI group-specific SNPs

Using the SnpEff software, the VCF files underwent annotation, allowing for the identification of SNPs specific to the high and low RFI groups. SnpSift v.5.1d was then employed to screen for SNP loci with significant functional and modifier-type impacts. This enabled the selection of candidate genes associated with these SNP loci.

## Gene function enrichment analysis and protein interaction network analysis

We employ clusterProfiler to conduct GO (Gene Ontology) and KEGG (Kyoto Encyclopedia of Genes and Genomes) enrichment analysis. The filter parameter is set as $p$ value < 0.05. The GO enrichment can further clarify the main biological functions of the genes where the specific SNPs are located. The KEGG pathway enrichment can be used to understand the signal pathway regulated by the genes. Using the STRING database (https://cn.string-db.org/), we perform protein interaction analysis on the relevant genes to select core genes that have interaction effects.

## Statistical analysis

The data from the experiment were analyzed and visualized using R v.4.3.0 (*R Core Team, 2023*) and Prism v.10.1.1 software. Statistical significance between the treatment and control groups was assessed using non-parametric tests or t-tests. A $p$-value of less than 0.05 was considered indicative of a significant difference. The corresponding code scripts can be found in Script S1.
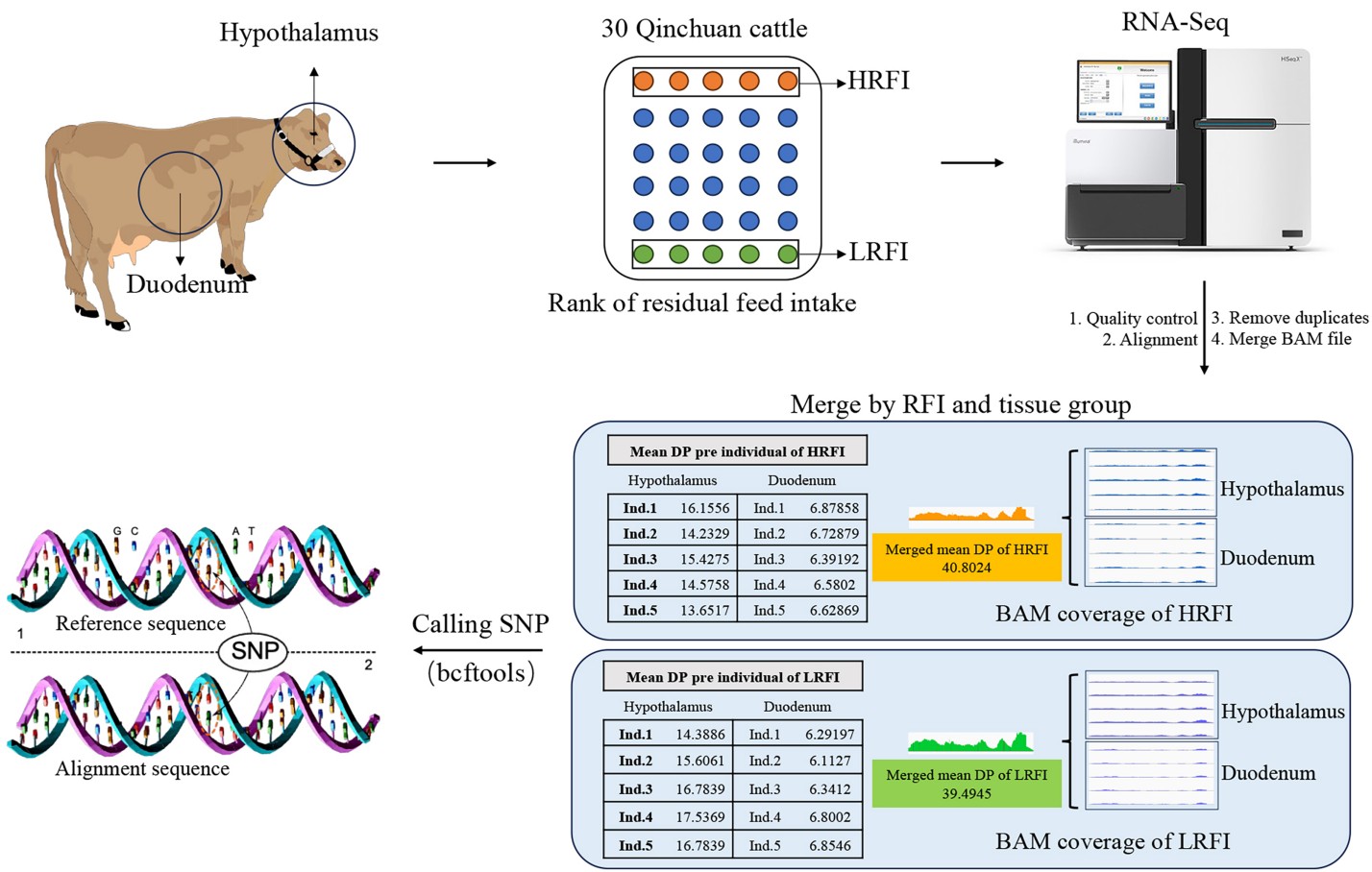

**Figure 1 Sample collection and bioinformatics analysis.** Hypothalamic and duodenal tissues from extreme RFI individuals were collected from 30 Qinchuan cows. After transcriptome sequencing, quality control, alignment, and deduplication, the alignment files were merged. The data from the two tissues were merged according to the RFI group, resulting in two alignment files. The merger greatly increased the depth (DP) of the reads, and the average reads DP of the two groups was basically consistent. Finally, the BCFtools software was used to identify SNPs in the merged data.

# RESULTS

## RNA-seq sequencing data quality and comparisons

From the RNA-seq data of the hypothalamus and duodenum, we obtained a total of 1,019 million and 275 million paired sequencing reads, respectively. After quality control, all 20 samples in this study had a Q30 score (error rate $p < 0.001$) above 96%. The GC% content was approximately 50% (Table S1). The clean data was aligned to the bovine reference genome, where the percentage of reads aligning to the reference genome was above 91%, and the percentage of reads with a unique alignment position ranged from 81.07% to 93.81% (Table S2). Analysis of the transcript expression for each sample indicated relatively consistent transcript abundances (Fig. 2). These results demonstrate the high quality of the obtained data, reducing the impact of sequencing errors on subsequent analysis.

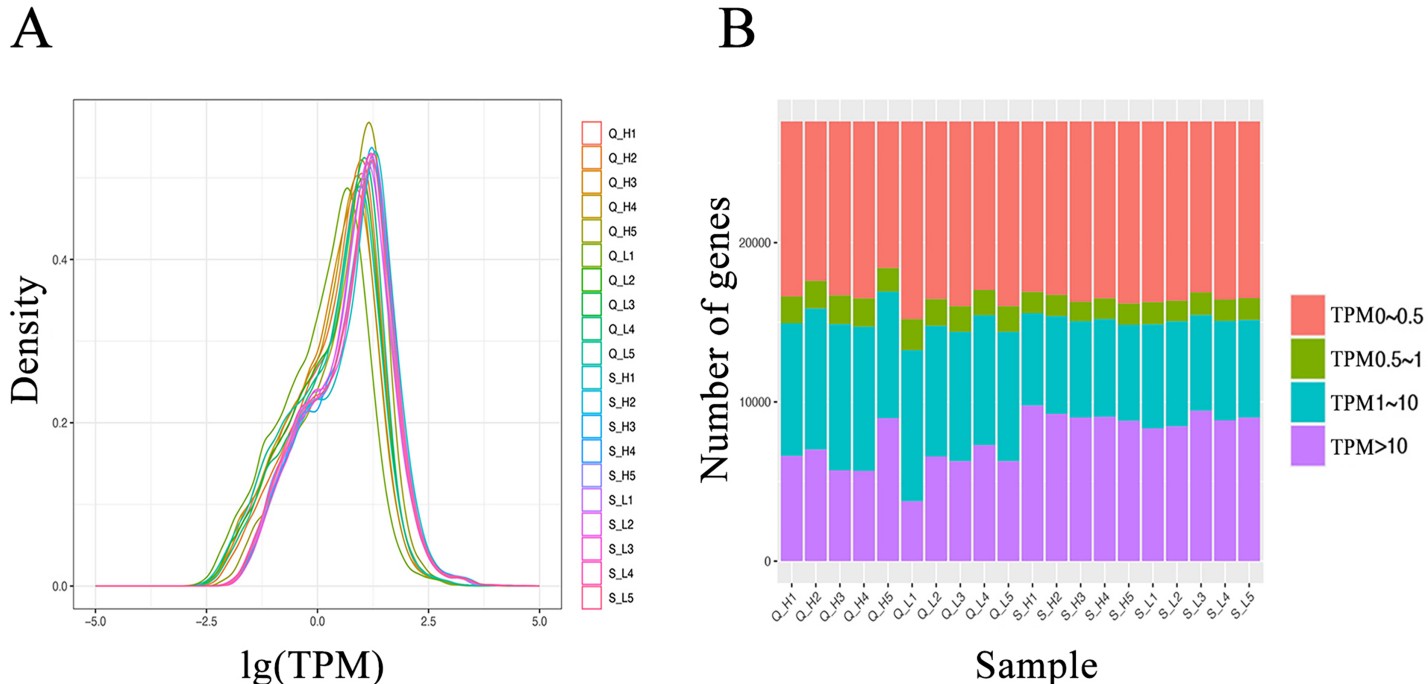

**Figure 2 Transcript expression (TPM) analysis of each sample.** (A) Transcript expression density in each sample. (B) Transcript expression in each sample.

**Table 1 Statistics on the number of homozygote and heterozygote SNPs in different RFI groups.**

| RFI group | SNP statistics | Homozygote sites | Heterozygote sites |
|---|---|---|---|
| HRFI | Number of SNPs | 11,991 | 258,419 |
| | Probability | 4.43% | 95.57% |
| LRFI | Number of SNPs | 14,007 | 241,113 |
| | Probability | 5.49% | 94.51% |

## SNPs screening and analysis

SNPs in the high and low RFI groups were identified by combining RFI phenotype with RNA-Seq data from hypothalamic and duodenal tissues. The numbers of homozygous and heterozygous mutations in the combined samples were counted (Table 1), revealing that there were 270,410 and 255,120 specific SNPs in the high and low RFI groups, respectively. Among these SNPs, 11,991 (4.43%) and 14,007 (5.49%) were homozygous mutations, with heterozygous mutations far outnumbering homozygous mutations. Statistical analysis based on the different types of mutations in the SNPs (Fig. 3) showed that the total number of transition types (A–G, C–T) was higher than the total number of transversion types (A–T, C–G, A–C, G–T). Among the six types of single nucleotide variations, A–G and C–T had the highest occurrence rates, while the occurrence rates of the other four types of transitions were relatively lower. In the high RFI group, A–G accounted for 36.5% and C–T accounted for 35.4%. In the low RFI group, A–G accounted for 35.7% and C-T accounted

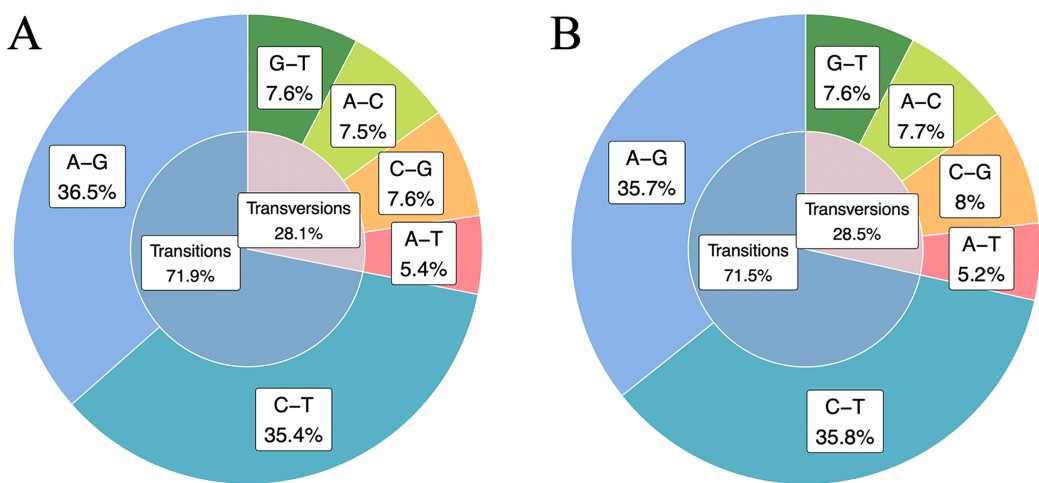

**Figure 3 High RFI group and low RFI group combined data SNPs type statistics.** (A) Statistics on SNPs types in the high RFI group. (B) Statistics on SNPs types in the low RFI group.

for 35.8%. The Ts/Tv ratios in the high and low RFI groups were 2.55 and 2.50, respectively. The differences in occurrence rates of the six types of single nucleotide variations between the high and low RFI groups were small, with a mean occurrence rate of transitions being 71.72% and transversions being 28.29%.

## Distribution statistics of SNPs

The study investigated the distribution and variation of SNPs across different chromosomes, providing insights into the genetic diversity among genes. Analysis of the combined high and low RFI groups revealed no significant difference ($p > 0.05$) in distribution between the two groups. Chromosome 1 exhibited the highest number of SNPs, while the mitochondria (MT) showed the lowest distribution (Fig. 4A). To account for differences in chromosome length, the ratio of SNPs number to chromosome length was calculated, revealing that the MT had the highest variant rate among both groups, indicating a higher density of SNPs per unit length (Fig. 4B). The chromosomal distribution of SNPs within the high and low RFI groups is delineated in Figs. 4C and 4D.

Comprehensive statistical analysis of SNP loci distribution across the genome was conducted for both high and low RFI datasets, emphasizing different genomic functional regions, such as downstream, exon, intergenic, intron, untranslated region, *etc*. (Table 2). The statistical analysis of the SNP locations in the genome for the high RFI and low RFI groups shows ten different distributions (DOWNSTREAM, EXON, INTERGENIC, INTRON, SPLICE_SITE_ACCEPTOR, SPLICE_SITE_DONOR, SPLICE_REGION, UPSTREAM, UTR_3_PRIME, UTR_5_PRIME) (Table 2). A single SNP may be located in multiple transcript regions. The analysis found that the INTRON region had the most SNP locations in both groups, with 429,995 areas annotated in the HRFI group and 435,881 areas in the LRFI group, significantly more than other functional regions. The next most abundant functional elements are INTERGENIC and DOWNSTREAM, while the remaining functional regions are less common. SPLICE_SITE_ACCEPTOR has the fewest

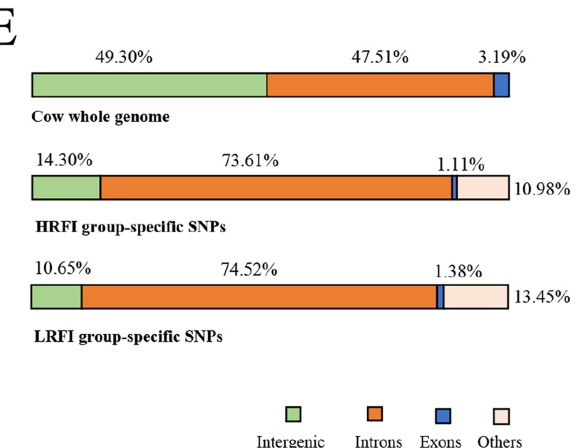

**A**

Number of variants (y-axis: 0, 5000, 10000, 15000) vs Chromosome (1–29, MT, X)

Legend: HRFI, LRFI

**B**

Variants rate (y-axis: 0, 5000, 10000, 15000, 20000) vs Chromosome (1–29, MT, X)

Legend: HRFI, LRFI

**C**

The number of SNPs within 1Mb window size

0Mb 18Mb 36Mb 54Mb 72Mb 90Mb 108Mb 126Mb 144Mb 159Mb

Chr1–Chr29, ChrMT, ChrX

Scale: 0, 128, 256, 384, 512, 640, 768, 896, 1024, 1152, 1280

**D**

The number of SNPs within 1Mb window size

0Mb 18Mb 36Mb 54Mb 72Mb 90Mb 108Mb 126Mb 144Mb 159Mb

Chr1–Chr29, ChrMT, ChrX

Scale: 0, 104, 208, 312, 416, 520, 624, 728, 832, 936, 1040

**E**

Cow whole genome: 49.30% / 47.51% / 3.19%

HRFI group-specific SNPs: 14.30% / 73.61% / 1.11% / 10.98%

LRFI group-specific SNPs: 10.65% / 74.52% / 1.38% / 13.45%

Legend: Intergenic, Introns, Exons, Others

**Figure 4 SNPs distribution statistics on chromosomes.** (A) Statistics on the number of SNPs on different chromosomes. (B) SNPs number and length ratio statistics on different chromosomes. (C) Distribution of SNPs on chromosome locations in the high RFI group. (D) Distribution of SNPs on chromosome locations in the low RFI group. (E) Percentage of the genome comprising each type of feature (top) and the proportion of SNPs detected by HRFI group-specific SNPs (middle) and LRFI group-specific SNPs (bottom) across these genomic features.

**Table 2 Genomic functional annotation of SNPs.**

|  | HRFI group | | LRFI group | |
|---|---|---|---|---|
|  | Count | Percent | Count | Percent |
| DOWNSTREAM | 35,350 | 6.05% | 43,281 | 7.40% |
| EXON | 6,504 | 1.11% | 8,070 | 1.38% |
| INTERGENIC | 83,530 | 14.30% | 62,282 | 10.65% |
| INTRON | 429,995 | 73.61% | 435,881 | 74.52% |
| SPLICE_SITE_ACCEPTOR | 37 | 0.01% | 73 | 0.01% |
| SPLICE_SITE_DONOR | 80 | 0.01% | 169 | 0.03% |
| SPLICE_SITE_REGION | 309 | 0.05% | 366 | 0.06% |
| UPSTREAM | 22,703 | 3.89% | 27,289 | 4.67% |
| UTR_3_PRIME | 4,371 | 0.75% | 5,920 | 1.01% |
| UTR_5_PRIME | 1,251 | 0.21% | 1,621 | 0.28% |

functional regions, with 37 in the high RFI group and 73 in the low RFI group. We hope to find most of the SNPs in the exonic regions, but coding regions generally experience higher selective pressure compared to non-coding regions (*Zhao et al., 2003*). The annotation of SNPs in the high and low RFI groups accounts for 1.11% and 1.38% in the exonic regions, respectively. At the same time, this also explains our detection results: the higher distribution of SNPs in intron regions is partly due to the fact that unspliced transcripts are also detected during sequencing, and partly because intron regions constitute 47.51% of the whole genome, which is significantly higher than the length of exonic regions (Fig. 4E). SNPs located in intergenic regions may be found in new genes or gene portions that have not been annotated yet.

## Influence prediction and amino acid change

By using the SnpEff software to evaluate the potential effects of SNP mutations on codons, it was found that over 98% of the SNPs in both groups were classified as modifiers, having minimal effect on genes and proteins. However, 159 SNPs in the high RFI group and 293 SNPs in the low RFI group were predicted to have a high effect (Figs. 5A, 5B; Tables S3, S4), warranting further investigation. This situation is as we expected, most SNPs are located in intron regions and intergenic areas, making it difficult to directly affect protein coding. Therefore, high-impact SNPs will be relatively fewer. Additionally, analyzing the overall levels of each amino acid can provide important insights into evolutionary pressures and adaptation mechanisms. This approach helps identify patterns and frequencies of amino acid substitutions in different biological contexts, enhancing our understanding of how these changes affect protein stability, function, and interactions. To analyze the potential impact of SNPs on genes and proteins, the effect of intergroup-specific SNPs (Tables S5, S6) on codons and subsequent amino acids was assessed. The analysis revealed that the amino acids most affected in both the high and low RFI groups were alanine-threonine, alanine-valine, and isoleucine-valine (Figs. 5C, 5D). By identifying SNP loci that significantly impacted both the high and low RFI groups and

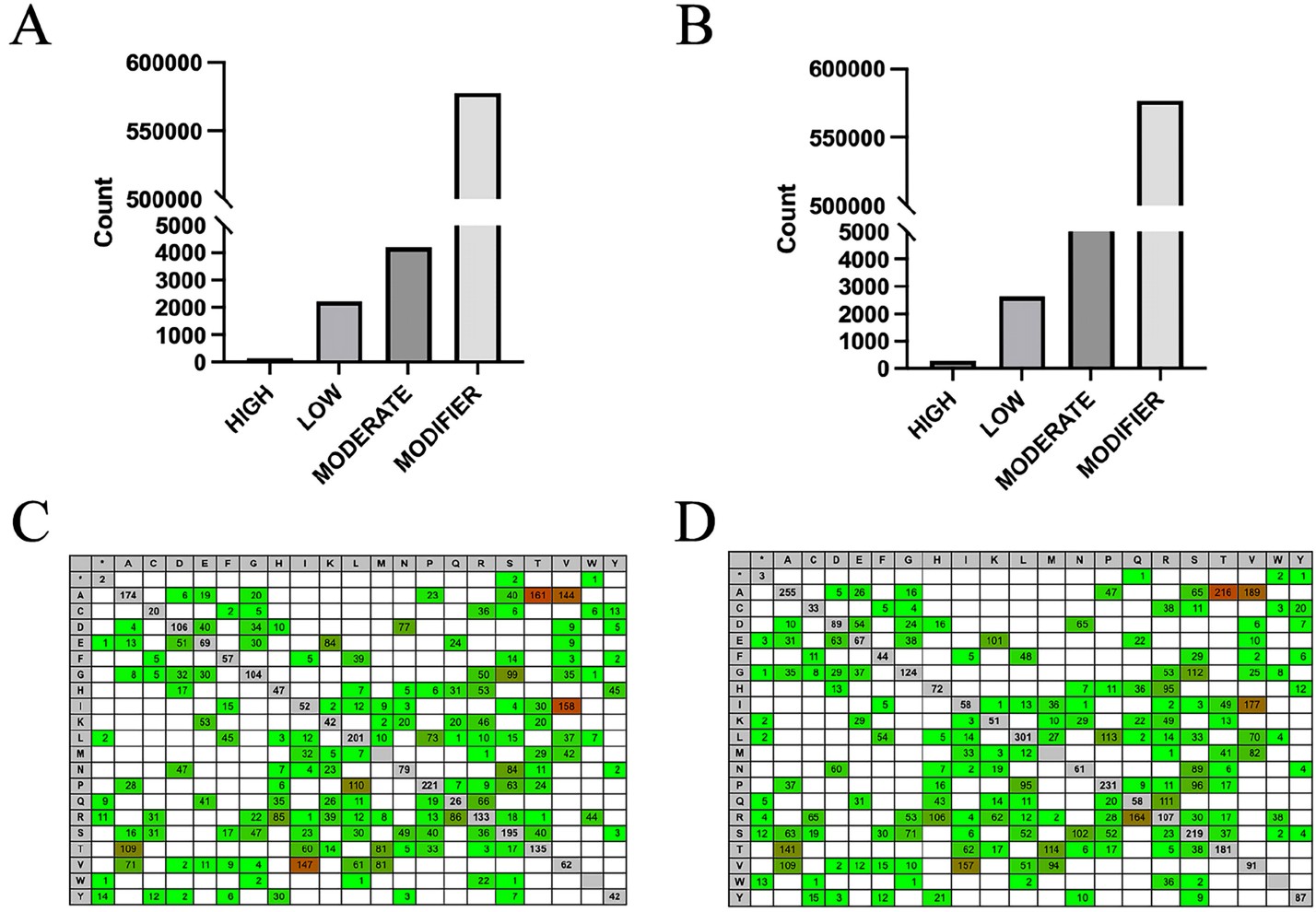

**Figure 5** **Influence prediction and amino acid change.** (A) High RFI group specific SNPs influence prediction statistics. (B) Low RFI group specific SNPs influence prediction statistics. (C) The amino acid changes caused by SNPs in the high RFI group were reference amino acids horizontally and altered amino acids vertically. Red background colors indicate that more changes happened (heat-map). (D) The amino acid changes caused by SNPs in the low RFI group.

mapping them to the corresponding genes using the SNPs annotation files, a total of 83 genes were identified in the high RFI group and 97 genes in the low RFI group. Interestingly, one gene, JSP.1, belonging to the MHC class I family, was common in both groups, and played a key role in regulating animal health within the immune system (*Hewitt, 2003*).

## Genes function annotation of high-impact SNP loci

GO functional annotation and enrichment analysis were conducted for the aforementioned genes (Figs. 6A, 6B). The results revealed that the enriched genes in the high and low RFI groups were primarily associated with protein binding and enzyme binding processes. Notably, a significant number of genes related to NADH activity were found in the low RFI (high feed efficiency) group. These genes were associated with oxidoreductase activity, acting on NADH or NADPH; NADH dehydrogenase activity;

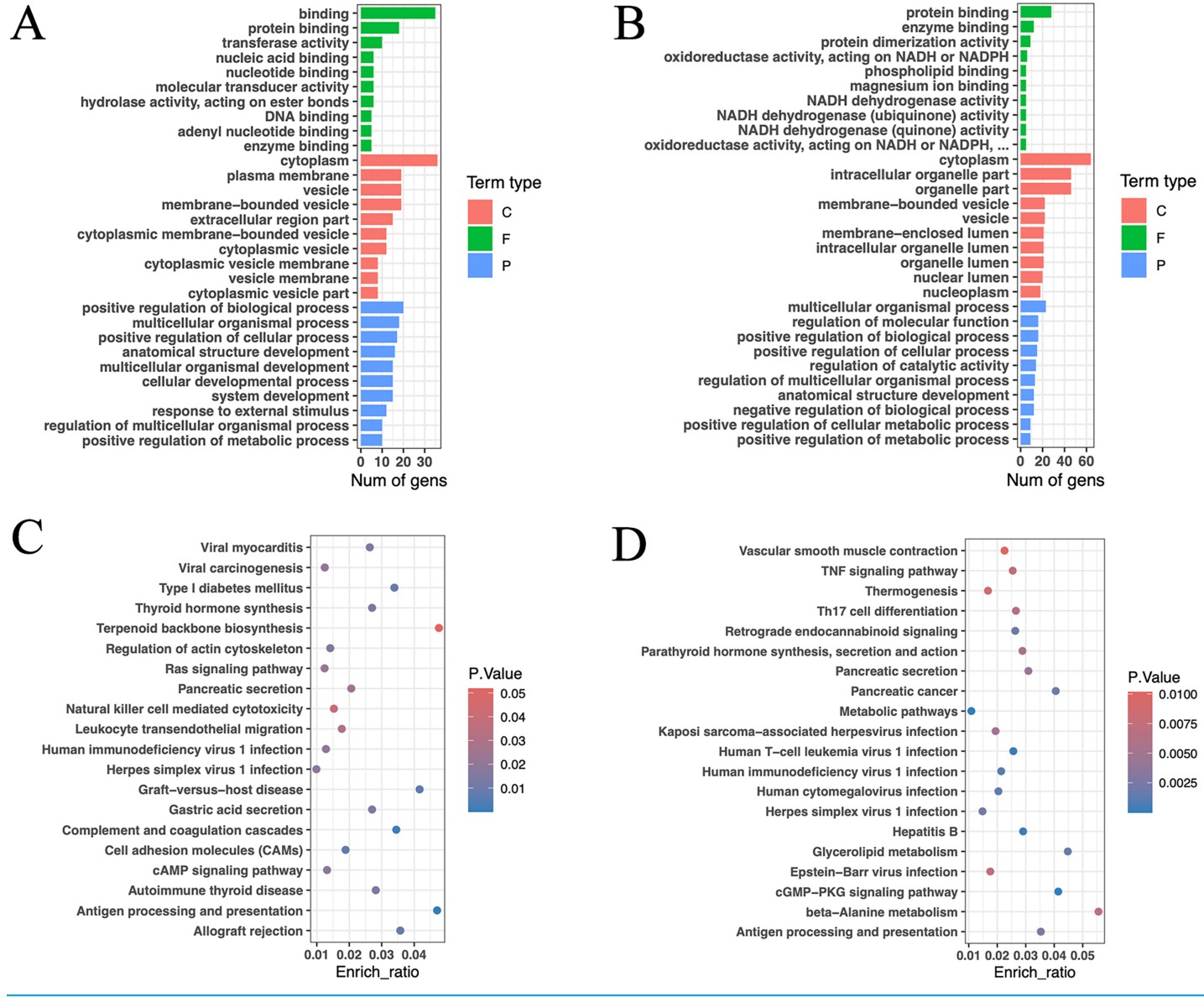

**Figure 6 Gene function annotation of high-impact SNP loci.** (A) Gene GO enrichment analysis (*p* < 0.05) of HRFI group-specific high-impact SNP sites, and select the top 10 for each term type based on *p*-value. (C: cellular component; F: molecular function; P: biological process). (B) Gene GO enrichment analysis (*p* < 0.05) of LRFI group-specific high-impact SNP sites, and select the top 10 for each term type based on *p* value. (C) Gene KEGG enrichment analysis (*p* < 0.05) of HRFI group-specific high-impact SNP sites, and select the top 20 based on *p* value. (D) Gene KEGG enrichment analysis (*p* < 0.05) of LRFI group-specific high-impact SNP sites, and select the top 20 based on *p* value.

NADH dehydrogenase (ubiquinone) activity; NADH dehydrogenase (quinone) activity; and oxidoreductase activity, acting on NADH or NADPH, quinone, or similar compounds as acceptors. NADH and its phosphorylation product NADPH play pivotal roles as coenzymes in various metabolic activities, including cell signaling, protein modification, energy metabolism, mitochondrial function, calcium homeostasis, antioxidative stress, biosynthesis, and cell death (*Berger, Ramirez-Hernandez & Ziegler, 2004*; *Patterson et al., 2005*; *Xiao et al., 2018*; *Ying, 2006*, *2007*, *2008*). In particular, the enrichment genes *ND4*,

*ND5*, and *ND6* are core subunits of the mitochondrial respiratory chain NADH dehydrogenase (complex I). They facilitate the transfer of electrons from NADH through the respiratory chain, utilizing ubiquinone as an electron acceptor, and are crucial for the catalysis and assembly of complex I (*UniProt, 2023*).

The KEGG enrichment analysis outcomes indicated enrichment of pathways related to thyroid hormone synthesis, pancreatic secretion, gastric acid secretion, cAMP signaling pathway, thermogenesis, parathyroid hormone synthesis and secretion, glycerolipid metabolism, TNF signaling pathway and beta-alanine metabolism in the high and low RFI groups (Figs. 6C, 6D). Notably, the thermogenesis pathway exhibited enrichment of *ND4*, *ND5*, and *ND6* genes. Additionally, *ATP1A2*, *SLC9A4*, and *PLA2G5* were identified as genes associated with energy metabolism (*Lingrel, 1992*; *Sakuta et al., 2020*; *Sun et al., 2004*).

## Protein-protein interaction analysis of high-impact SNP loci

Protein-protein interaction analysis is a method used to study the interactions between proteins, which can be employed to uncover the relationships and networks among proteins, consequently explaining functional interactions and illustrating the intricate interconnections between proteins. Our results revealed distinct patterns of core genes and interaction relationships between the two groups. In the high RFI group, we identified 29 core genes and 23 interaction relationships, while in the low RFI group, we found 42 core genes and 41 interaction relationships (Fig. 7). Several genes, such as *HSP90AA1*, *EIF2AK3*, *PAK1*, *MAP3K7*, *PGM2L1*, *DNM1L* and *CYB5R3*, were found to be related to energy metabolism, fat deposition and muscle development (*Badri et al., 2018*; *Charoensook et al., 2012*; *Chen et al., 2019*; *Chiang & Jin, 2014*; *Hogarth et al., 2018*; *Liu et al., 2024*; *Lopez-Bellon et al., 2022*; *Zhang, O'Keefe & Jonason, 2017*).

## Analysis of candidate gene SNP loci

Based on the results of GO and KEGG analysis, we focused on phenotype-related terms. In the high RFI group we screened GO terms: positive regulation of metabolic process and multicellular organismal development; KEGG terms: thyroid hormone synthesis, gastric acid secretion, cAMP signaling pathway, pancreas signaling pathway. In the low RFI group we screened GO terms: oxidoreductase activity, acting on NADH or NADPH; KEGG terms: thermogenesis, metabolic pathways, TNF signaling pathway. Finally, 18 genes were identified in the high RFI group and 21 genes in the LRFI group (Tables S3, S4, S7). Also combining protein interaction analysis and existing studies, we finally focused on 14 genes. In these genes, we combined the prediction results of SNP impact, showing high-impact SNP sites in the genes. We found that these SNPs are mostly located in the exon regions, and most are A-G mutation types. This type of mutation might change the coded amino acids, and affect the structure and function of the protein (Table 3). However, their specific function and roles would be detected in the future studies for clarifying their variant effect to phenotype.

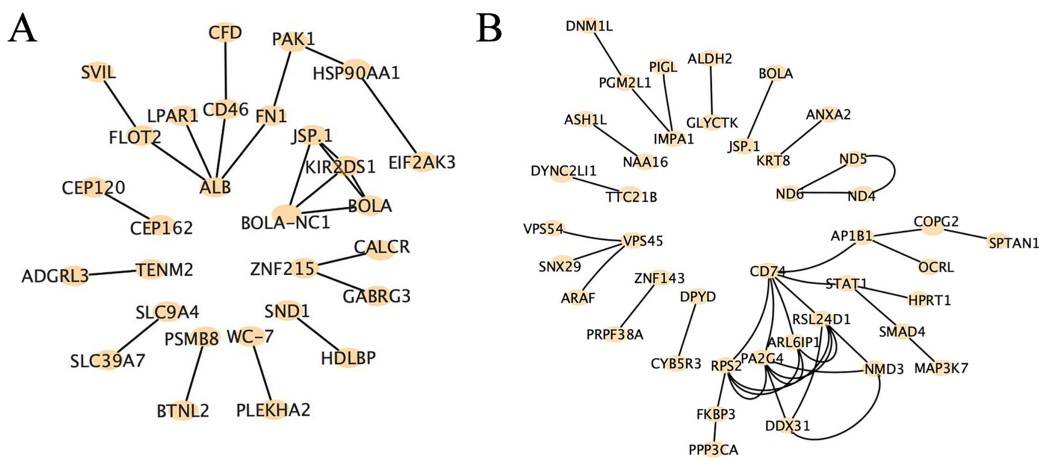

**Figure 7 Protein-protein interaction analysis of high-impact SNP loci.** (A) Protein-protein interaction analysis of high-impact SNP sites in HRFI group, using string database, composed of 29 nodes and 23 edges. (B) Protein-protein interaction analysis of high-impact SNP sites in LRFI group, using string database, composed of 42 nodes and 41 edges.

**Table 3 Location of SNPs from candidate genes.**

| Gene | Chromosome | Site | Reference nucleotide | Mutated nucleotide | Group | References |
|---|---|---|---|---|---|---|
| ATP1A2 | 3 | 9,540,257 | C | G | High RFI | *Lingrel (1992)* |
| SLC9A4 | 11 | 7,280,369 | G | A | High RFI | *Sakuta et al. (2020)* |
| PLA2G5 | 2 | 132,645,789 | G | A | High RFI | *Sun et al. (2004)* |
| HSP90AA1 | 21 | 66,941,687 | C | T | High RFI | *Badri et al. (2018)*, *Charoensook et al. (2012)* |
| EIF2AK3 | 11 | 47,484,569 | G | A | High RFI | *Chen et al. (2019)* |
| PAK1 | 29 | 18,525,245 | A | T | High RFI | *Chiang & Jin (2014)* |
| MAP3K7 | 9 | 59,698,567 | A | T | Low RFI | *Zhang, O'Keefe & Jonason (2017)* |
| PGM2L1 | 15 | 53,718,210 | A | G | Low RFI | *Liu et al. (2024)* |
| DNM1L | 5 | 77,174,829 | A | C | Low RFI | *Hogarth et al. (2018)* |
| CYB5R3 | 5 | 113,485,291 | T | G | Low RFI | *Lopez-Bellon et al. (2022)* |
| SMAD4 | 24 | 50,564,735 | G | A | Low RFI | *Li et al. (2020)* |
| ND4 | MT | 11,329 | A | G | Low RFI | *Yang et al. (2023)* |
| ND5 | MT | 12,672 | G | A | Low RFI | *Yang et al. (2023)* |
| ND6 | MT | 14,066 | C | T | Low RFI | *Yang et al. (2023)* |

# DISCUSSION

Enhancing animal feed efficienvy and reducing production costs are key objectives in livestock production, as they are essential for animal growth and development. Besides development high-quality feeds through selective breeding, investigating genetic factors affecting feed efficiency in beef cattle is a significant research focus. The hypothalamus acts as a central control regulator of feeding, interacting with neuron groups to produce signals that stimulate or suppress appetite, thereby influencing food intake (*Perkins et al., 2014*; *Sartin, Whitlock & Daniel, 2011*). The duodenum, the initial segment of the small intestine,

is crucial for digestion and absorbing nutrients, particularly carbohydrates and micronutrients (*Cooke & Clark, 1976*; *Reeves & Chaney, 2004*). The brain-gut axis, comprising the central, enteric, and autonomic nervous systems, facilitates the complex communication between the gut and the brain *via* neurohumoral pathways (*Margolis, Cryan & Mayer, 2021*). Axes like the hypothalamic-pituitary-adrenal (HPA) axis, part of the brain-gut axis, are identified as key contributors to the variability in RFI (*DiGiacomo et al., 2018*).

RNA-Seq, a second-generation transcriptome sequencing technology, offers various approaches approaches for high-throughput functional genomics, including gene expression profiling (*Song et al., 2019*), genome annotation (*Li et al., 2011*), non-coding RNA discovery (*Jiang et al., 2022*), and gene mutation analysis (*Lopez-Maestre et al., 2016*). These approaches collectively elucidate the intricate complexities biological systems. Residual feed intake, a quantitative trait, is governed by multiple genes and influenced by diverse physiological metabolic processes (*Arthur et al., 2001*). Previous research has effectively identified genes linked to feed utilization efficiency through SNP screening. For instance, *Higgins et al. (2018)* found a strong association between the variant rs43555985 and RFI ($P = 8.28E{-}06$). *Bolormaa et al. (2011)* identified 111 and 75 significantly associated SNPs with RFI ($p < 0.001$) using the 10 K and 50 K SNP microarray data, respectively. *Lima et al. (2016)* used comprehensive GWA, AWM, and RNA-Seq analyses to identify the *PRUNE2* gene as a potential candidate affecting feed efficiency. Various tools have been developed for SNP detection from RNA-seq data and for determining concordance of SNP and genotype detection between RNA-seq and DNA-seq (*Dobin et al., 2013*; *Liu, Shen & Bao, 2022*; *Luo et al., 2019*; *Quinn et al., 2013*; *Tang et al., 2014*; *Van der Auwera et al., 2013*). In our study, we collected hypothalamic and duodenal tissues from beef cattle with high and low RFI. Using high-throughput transcriptome sequencing, we obtained data from 20 samples and merged the tissue data by RFI groups to identify SNPs, aiming to improve the accuracy of SNP functional annotation.

SNPs represent DNA sequence variations due to single nucleotide changes. Analysis SNPs data from 20 samples revealed 270,410 unique SNPs in the high RFI group and 255,120 in the low RFI group. The high RFI group had 11,991 homozygous SNPs, while the low RFI group had 14,007. Over 70% of SNPs in both groups were located in the intron region, followed by the intergenic region, likely due to unspliced transcripts (premature transcripts) and unannotated regions (*Jehl et al., 2021*). However, only 1.11% and 1.38% of SNPs in high and low RFI groups were in the intron region. This distribution is anticipated, as intronic regions typically face higher selection pressures compared to other non-coding regions (*Zhao et al., 2003*). Furthermore, the transition-to-transversion ratio (Ts/Tv) was of 2.55 in the high RFI group and 2.50 in the low RFI group, reflecting a higher incidence of transition mutations, consistent with previous findings (*Nandanpawar et al., 2023*; *Raizada & Souframanien, 2019*; *Van Deventer et al., 2020*). The consistency of Ts/Tv values supports the reliability of SNP identification in this study (*Arabnejad et al., 2018*). Using SnpEff software for SNP annotations, functionally significant SNPs unique to the high and low RFI groups were identified, along with their corresponding genes. A total of

83 genes were found in the high RFI group, while 97 genes were identified in the low RFI group.

GO and KEGG pathway enrichment analyses were conducted independently for each gene set. At the molecular function level, the enriched GO terms in both groups were primarily associated with protein binding and enzyme binding. In the low RFI group, specific enrichment was noted for NADH-related terms, including oxidoreductase activity acting on NADH or NADPH, NADH dehydrogenase activity, NADH dehydrogenase (ubiquinone) activity, NADH dehydrogenase (quinone) activity, and oxidoreductase activity acting on NADH or NADPH, quinone or similar compound as acceptor. NADH dehydrogenase, also known as NADH: ubiquinone oxidoreductase or complex I, facilitating electron transfer from NADH to coenzyme Q, crucial for energy metabolism in the mitochondrial inner membran (*Nakamaru-Ogiso et al., 2010*). The genes *ND4*, *ND5*, and *ND6* are core subunits of mitochondrial respiratory chain NADH dehydrogenase (complex I), essential for its catalytic function and assembly (*UniProt, 2023*). Previous genomic analyses have suggested that the *ND (2,3,4,4L,5,6)* gene cluster may significantly impact feed efficiency changes (*Yang et al., 2023*). Mitochondria generate approximately 90% of cellular energy and are abundant in metabolically active cells, such as liver, kidney, muscle, and brain cells. Studies in poultry and livestock have shown a close relationship between feed efficiency and mitochondrial function and biochemistry. Research indicated that animals with low RFI exhibit increased rates of mitochondrial respiration (*Kolath et al., 2006*), enhanced coupling of the electron transport chain (*Bottje & Carstens, 2009*), higher activity of respiratory chain complexes I-V (*Iqbal et al., 2005*), and lower heat production per kilogram of metabolic body weight (MBW) (*Nkrumah et al., 2006*). Moreover, the electron transport chain is also recognized as the site of reactive oxygen species (ROS) production, and elevated ROS levels pose a significant threat to the antioxidant defense system by increasing the susceptibility of various cellular components to oxidative damage (*Nolfi-Donegan, Braganza & Shiva, 2020*). Animals with higher feed efficiency tend to exhibit lower oxidative stress phenomena (*Bottje & Carstens, 2009*; *Iqbal et al., 2005*, *2004*). KEGG pathway enrichment analysis also identified several pathways associated with energy metabolism. In the high RFI group, enriched pathways included thyroid hormone synthesis, pancreatic secretion, gastric acid secretion, and cAMP signaling pathway were enriched. Enriched pathways in the low RFI group included thermogenesis, parathyroid hormone synthesis, secretion and action, pancreatic secretion, triglyceride metabolism, and alanine metabolism.

One of the enriched genes, *ATP1A2*, is involved in ATP hydrolysis and facilitates sodium and potassium ion exchange across the plasma membrane, establishing an electrochemical gradient for the active transport of nutrients (*Lingrel, 1992*). Additionally, *SLC9A4* functions act as a sodium ion sensor, regulating water intake behavior (*Sakuta et al., 2020*). *PLA2G5* is speculated to play a role in the biosynthesis of N-acyl ethanolamines, compounds that regulate energy metabolism (*Sun et al., 2004*). Through protein-protein interaction analysis, we have discovered that genes such as *HSP90AA1*, *EIF2AK3*, *PAK1*, *SMAD4*, *MAP3K7*, *PGM2L1*, *DNM1L*, and *CYB5R3* were found to be related to metabolism, which might be related to cattle RFI variants. For example, genetic
variations in *HSP90AA1* are associated with thermoregulatory traits in cattle (*Badri et al., 2018*; *Charoensook et al., 2012*). Activation of *EIF2AK3* has been shown to promote metabolic dysfunctions (*Chen et al., 2019*). *PAK1* is involved in the regulation of glucose uptake (*Chiang & Jin, 2014*). *SMAD4* is linked to aerobic glycolysis and obesity (*Li et al., 2020*). *MAP3K7* induces adipocyte differentiation *via PPARγ* signaling (*Zhang, O'Keefe & Jonason, 2017*). *PGM2L1* is suggested to be related to meat quality and muscle development in sheep (*Liu et al., 2024*). Variations in *DNM1L* can lead to mitochondrial fragmentation, decreased membrane potential, reduced oxidative capacity, and increased levels of reactive oxygen species (ROS) (*Hogarth et al., 2018*). *CYB5R3* works with coenzyme Q, participating in the cross-membrane redox system to protect cells against oxidative stress (*Lopez-Bellon et al., 2022*). These genes will be targeted in future cellular and molecular experiments to validate their associations with the RFI trait.

RFI, an essential economic trait in feed efficiency research, necessitates a thorough understanding of the genetic mechanisms associated with SNP loci and their impact on RFI regulation in beef cattle. This understanding is vital for analyzing RFI variation in livestock and improving feed conversion efficiency for sustainable and cost-effective animal husbandry. Additionally, enhancing feed utilization efficiency can reduce methane emissions, improve animal health and production performance, and serve as a foundation for selecting and breeding feed-efficient beef cattle. The use of transcriptome data to identify SNPs in the investigation of RFI offers several distinct advantages. Firstly, transcriptome data provides valuable insights into gene expression, thereby enabling researchers to focus directly on genes and SNPs that are associated with specific physiological processes (*Jehl et al., 2021*). Secondly, the acquisition of transcriptome data is relatively cost-effective, particularly in species for which a reference genome is not available (*Lopez-Maestre et al., 2016*). Furthermore, transcriptome data can be integrated with genomic data, leveraging methodologies such as genome-wide association studies (GWAS) and expression quantitative trait locus (eQTL) analyses to further validate the associations between SNPs and RFI (*Ibragimov et al., 2022*). By optimizing the analytical workflows for RNA-Seq data, the accuracy and reliability of SNP detection can be significantly enhanced, thereby reducing the incidence of false positives. Additionally, transcriptome data can be employed to validate SNPs identified from genomic data, thereby augmenting the reliability of the results (*Ge, Li & Zhang, 2024*; *Lam et al., 2020*).

Although we have obtained many SNPs that are meaningful and may have critical genetic effects. However, given many drawbacks of RNA-Seq, such as the uneven depth distribution of reads across the genome from RNA-Seq data (*Jehl et al., 2021*), the large variation in RNA expression levels in different tissues, cells, and physiological stages (*Sims et al., 2014*), and the fact that the variants detected at the RNA level may not exist at the DAN level (*Jehl et al., 2021*), and the fact that the SNPs detection near exon-exon junctions still needs to remain cautious (*Lagarrigue et al., 2013*; *Peng et al., 2012*). SNPs analysis from RNA-seq data should also continuously improve its identification efficiency, or combine with other methods such as sanger sequencing, flight mass spectrometry, fluorescent probes and so on to improve the recognition rate and accuracy of valid SNPs, so as to effectively improve the efficiency of the identification of SNPs in the coding region, to

increase the reliability of the data results, and to reduce false positives. This study also has certain limitations. SnpEff primarily relies on annotation databases and predefined rules to predict the functional impact of SNPs. While this provides a quick assessment, it lacks the comprehensive validation offered by multiple bioinformatics tools. For instance, tools like SIFT (*Ng & Henikoff, 2003*), PolyPhen2 (*Adzhubei et al., 2010*), and Panther (*Tang & Thomas, 2016*) can provide more detailed functional impact predictions. The use of a single tool may lead to partial and inaccurate predictions. Moreover, directly provide predictions on protein stability changes is significative. Tools such as I-Mutant (*Capriotti, Fariselli & Casadio, 2005*), Mupro (*Laskar et al., 2023*), and CUPSAT (*Parthiban, Gromiha & Schomburg, 2006*) can predict the impact of SNPs on protein stability by calculating changes in free energy ($\Delta\Delta G$), which is crucial for understanding how mutations affect protein folding and function. Molecular dynamics simulations can offer detailed information on how mutations impact protein structure and dynamics, revealing conformational changes and functional effects (*Elangeeb et al., 2024*; *Kamal et al., 2024*). To achieve more comprehensive and accurate SNP functional predictions, it is recommended to integrate multiple bioinformatics tools and methods. This approach will enhance the accuracy and reliability of the research findings.

## CONCLUSIONS

Due to its low cost and effective detection, RNA-Seq data has become a reliable resource for polymorphism detection in non-model animals. In this study, RFI-related SNPs and their annotated genes were obtained by integrating multiple tissue RNA-seq data from extreme RFI individuals to improve SNP identification. Variants calling based on RNA-seq data can effectively improve the identification of phenotype-related SNPs, which is an efficient and feasible approach to get potential functional SNPs. By mining SNPs with high impact on genes, this genes and SNPs related to RFI would be helpful and valuable for molecular validation in subsequent studies.

## ACKNOWLEDGEMENTS

We thank all the cattle breeding staff who helped with sample collecting in this experiment and all members of this research group for all valuable discussions and feedback on this article.

### Funding

This work was supported by the National Natural Science Foundation of China (32160776), a grant from the China Agriculture Research System (CARS-36), the Natural Science Foundation of Ningxia Province, China (Grant No. 2021AAC03027), and West Light Foundation of the Chinese Academy of Sciences (Grant No. XAB2022YW11). The funders had no role in study design, data collection and analysis, decision to publish, or preparation of the manuscript.

## Grant Disclosures

The following grant information was disclosed by the authors:
National Natural Science Foundation of China: 32160776.
China Agriculture Research System (CARS-36).
Natural Science Foundation of Ningxia Province, China: 2021AAC03027.
and West Light Foundation of the Chinese Academy of Sciences: XAB2022YW11.

## Competing Interests

Cong-Jun Li is an Academic Editor for PeerJ.

## Author Contributions

- Zonghua Su performed the experiments, analyzed the data, prepared figures and/or tables, and approved the final draft.
- Chenglong Li performed the experiments, analyzed the data, prepared figures and/or tables, and approved the final draft.
- Chaoyun Yang performed the experiments, prepared figures and/or tables, and approved the final draft.
- YanLing Ding performed the experiments, prepared figures and/or tables, and approved the final draft.
- Xiaonan Zhou performed the experiments, prepared figures and/or tables, and approved the final draft.
- Junjie Xu analyzed the data, prepared figures and/or tables, and approved the final draft.
- Chang Qu analyzed the data, prepared figures and/or tables, and approved the final draft.
- Yuangang Shi conceived and designed the experiments, authored or reviewed drafts of the article, and approved the final draft.
- Cong-Jun Li conceived and designed the experiments, authored or reviewed drafts of the article, and approved the final draft.
- Xiaolong Kang conceived and designed the experiments, authored or reviewed drafts of the article, and approved the final draft.

## Animal Ethics

The following information was supplied relating to ethical approvals (*i.e.*, approving body and any reference numbers):

All experimental animal studies were reviewed and ap- proved by the Animal Welfare Committee of Ningxia University (permit number NXUC20211015).

## DNA Deposition

The following information was supplied regarding the deposition of DNA sequences:
The duodenal tissue data is available at ENA: PRJNA747740.

## Data Availability

The code is available in the Supplemental File.

## Supplemental Information

Supplemental information for this article can be found online at http://dx.doi.org/10.7717/peerj.19270#supplemental-information.

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
