# Peer review of "Identification of single nucleotide polymorphisms (SNPs) potentially associated with residual feed intake in Qinchuan beef cattle by hypothalamus and duodenum RNA-Seq data"

_PeerJ, doi:10.7717/peerj.19270_

## Round 0.1 · original submission · Major Revisions

While the reviewers agreed that the manuscript was well written, there were still some concerns around methodological details and areas where clarity could be improved.

Reviewer 1 ·

Basic reporting

o The manuscript is written in professional and clear English. However, there are some long and complex sentences, particularly in the introduction and discussion sections, which may benefit from simplification for broader accessibility.
o Examples of cumbersome phrases: "These variants can potentially impact gene expression and function, consequently influencing individual phenotypes and disease susceptibilities" (Introduction). Simplifying could improve readability.
o The introduction provides sufficient context regarding the significance of residual feed intake (RFI) in cattle and its economic and environmental importance. References are comprehensive and relevant.
o The manuscript conforms to PeerJ standards. Figures are relevant and well-labeled, though Figure 5 could be better explained in the text to improve comprehension of its relationship to SNP impacts.
o Supplementary tables and figures are detailed and contribute to the manuscript's overall clarity.
o Raw sequencing data is described but requires a verification check for completeness, accessibility, and adherence to deposition policies.

Experimental design

o The research aligns well with the journal's scope. It addresses a defined knowledge gap by associating hypothalamus and duodenum SNPs with RFI in beef cattle.
o The experimental design is robust, with detailed descriptions of RNA extraction, sequencing, quality control, SNP filtering, and bioinformatics analyses. The inclusion of statistical power calculations enhances the methodological rigor.
o Ethical considerations are explicitly addressed, with approvals cited.
o The methods are described with sufficient detail to allow replication, but the manuscript could include more specifics about SNP prioritization criteria for functional studies.

Validity of the findings

o The sequencing data quality is strong, with high alignment rates to the bovine genome and consistent transcript expression levels. SNP identification is thorough, supported by statistical validation.
o The discussion effectively links SNP findings to known pathways like NADH-related metabolism and energy efficiency, providing plausible biological mechanisms. However, claims regarding the impact of specific SNPs on energy metabolism require experimental validation.
o Conclusions are well-supported by the results, although the authors should emphasize the exploratory nature of SNP associations and recommend further experimental studies.

Additional comments

o The manuscript could benefit from discussing limitations more explicitly, such as the challenges of SNP detection in RNA-Seq data due to technical biases.
o Author may include recent paper on in silico application "Comprehensive in silico analysis of prolactin receptor (PRLR) gene nonsynonymous single nucleotide polymorphisms (nsSNPs) reveals multifaceted impact on protein structure, function, and interactions" (https://doi.org/10.1080/07391102.2024.2335295) and other recent papers.
o While protein-protein interaction networks are explored, their direct link to the RFI phenotype remains speculative.
o Simplify and clarify some complex sentences.
o Expand on the experimental validation of SNPs with high predicted impacts.
o Include a more detailed discussion on potential biases in RNA-Seq-based SNP identification.

Reviewer 2 ·

Basic reporting

The manuscript entitled “Identification of SNPs potentially associated with residual feed intake in beef cattle by hypothalamus and duodenum RNA-Seq data” is now reviewed. The manuscript is written well, on an interesting topic which will be of great importance for researchers working in similar field. Overall, the topic is well introduced and a good amount of bioinformatic analysis was carried out on RNA-seq data of Qinchuan beef cattle (n=30) to generate the results output.

The manuscript is written in clear, unambiguous and professional English. Sufficient references are cited and background information is sufficient.

The manuscript needs only minor revision on a few points before it should be accepted for publication. These are mentioned below in the attached file.

Experimental design

The research question is well defined, relevant and meaningful. Most of the methods are sufficient, however, a few methods need more details which are mentioned in the attached file.

Validity of the findings

The study is important. However, the statistical test used in the study may be more robust, and the same is mentioned how to improve the statistical analysis.

The conclusion may be revised, focussing on the major findings of the study, rather than using some general statements.

Additional comments

The details are given in the attached file.

Annotated reviews are not available for download in order to protect the identity of reviewers who chose to remain anonymous.

·

Basic reporting

no comment

Experimental design

the method is detailed and clear but needs improvement for the samples and the feeding given.

Validity of the findings

no comment

Additional comments

this research is very good to support the development of feed-efficient livestock from the genetic field.

---

## Round 0.2 · accepted · Accept

Both reviewers were satisfied that the revisions made had addressed their previous concerns, and the manuscript is now ready for publication.

Reviewer 1 ·

Basic reporting

No comment

Experimental design

No comment

Validity of the findings

No comment

Additional comments

No comment

Reviewer 2 ·

Basic reporting

The revised manuscript "Identification of SNPs potentially associated with residual feed intake in Qinchuan beef cattle by hypothalamus and duodenum RNA-Seq data" is now reviewed. The manuscript is now much improved and the authors have addressed all the issues raised in the earlier round of review satisfactorily.

The manuscript is written in a clear and unambiguous professional English, with sufficient background information and up-to-date literature references.

Experimental design

The experimental design is sound and the primary research is within the aims and scope of the journal. The research question is well-defined, relevant and meaningful for the readers working in this area.

Validity of the findings

All the underlying data have been provided, they are robust and statistically sound, controlled and the validity of the findings is clearly linked to the original research question.

The conclusion is now well stated and limited to the supporting results.

Additional comments

The revised manuscript is now suitable for publication in a reputed journal like PeerJ.